# Syntenic Cell Wall QTLs as Versatile Breeding Tools: Intraspecific Allelic Variability and Predictability of Biomass Quality Loci in Target Plant Species

**DOI:** 10.3390/plants12040779

**Published:** 2023-02-09

**Authors:** Francesco Pancaldi, Eibertus N. van Loo, Sylwia Senio, Mohamad Al Hassan, Kasper van der Cruijsen, Maria-João Paulo, Oene Dolstra, M. Eric Schranz, Luisa M. Trindade

**Affiliations:** 1Plant Breeding, Wageningen University & Research, Droevendaalsesteeg 1, 6708 PB Wageningen, The Netherlands; 2Biometris, Wageningen University & Research, Droevendaalsesteeg 1, 6708 PB Wageningen, The Netherlands; 3Biosystematics, Wageningen University & Research, Droevendaalsesteeg 1, 6708 PB Wageningen, The Netherlands

**Keywords:** QTL analysis, GWAS, biomass quality, cell wall, synteny, genomics, miscanthus, switchgrass

## Abstract

Syntenic cell wall QTLs (SQTLs) can identify genetic determinants of biomass traits in understudied species based on results from model crops. However, their effective use in plant breeding requires SQTLs to display intraspecific allelic variability and to predict causative loci in other populations/species than the ones used for SQTLs identification. In this study, genome assemblies from different accessions of Arabidopsis, rapeseed, tomato, rice, Brachypodium and maize were used to evaluate the intraspecific variability of SQTLs. In parallel, a genome-wide association study (GWAS) on cell wall quality traits was performed in miscanthus to verify the colocalization between GWAS loci and miscanthus SQTLs. Finally, an analogous approach was applied on a set of switchgrass cell wall QTLs retrieved from the literature. These analyses revealed large SQTLs intraspecific genetic variability, ranging from presence–absence gene variation to SNPs/INDELs and changes in coded proteins. Cell wall genes displaying gene dosage regulation, such as PAL and CAD, displayed presence–absence variation in Brachypodium and rapeseed, while protein INDELs were detected for the Brachypodium homologs of the rice brittle culm-like 8 locus, which may likely impact cell wall quality. Furthermore, SQTLs significantly colocalized with the miscanthus and switchgrass QTLs, with relevant cell wall genes being retained in colocalizing regions. Overall, SQTLs are useful tools to screen germplasm for relevant genes and alleles to improve biomass quality and can increase the efficiency of plant breeding in understudied biomass crops.

## 1. Introduction

The projection of known quantitative trait loci (QTLs) across species through genome synteny was recently shown to allow for the quick identification of conserved genomic regions underlying traits of interest in large panels of plants, including novel, under-domesticated, crops [1]. On the one hand, this is possible thanks to the established relationship between the occurrence of gene synteny (i.e., the conservation of gene presence and gene order across genomes) and the conservation of gene function across diverse living organisms [2,3,4]. On the other hand, the colocalization of syntenic regions with previously mapped QTLs ensures the relevance of the regions identified for the involvement in the trait of interest, since the occurrence of QTLs proves that genomic regions are causative of trait variability in particular species. The genomic regions identified across several species by combining synteny and QTL information can be referred to as syntenic quantitative trait loci (SQTLs) [1].

The concept of SQTLs was demonstrated for plant cell wall compositional traits [1]. These traits entail the relative amounts and the chemical–physical properties of the polysaccharides that constitute plant cell walls and underly feedstock quality of biomass crops—mainly cellulose, hemicelluloses, pectins, and lignin [5,6]. The generally high syntenic conservation of cell wall genes within previously mapped cell wall QTLs allowed for the identification of numerous SQTLs, highlighting the potential of this strategy for projecting important conserved loci underlying biomass quality across multiple plants [1]. Furthermore, cell wall SQTLs were shown to allow for a “fine-mapping” of the initial cell wall QTLs, by assessing the overlap of multiple initial QTLs on specific syntenic segments across multiple genomes [1]. Finally, cell wall SQTLs contain relevant cell wall genes, known from previous studies to affect cell wall quality in different species, and therefore likely representing (some of) the conserved causative genes of the initial cell wall QTLs [1].

What was just discussed demonstrates that SQTLs represent valuable tools during the pre-breeding steps of crop improvement. Specifically, their availability, combined with the dropping of genome sequencing costs, allows for the potential circumvention (of part of) the pre-breeding studies on trait genetics needed to start breeding programs in under-domesticated crops [1]. Given the wealth of genetic resources represented by under-domesticated plant species [7], the availability of tools to speed up their improvement is an extremely important asset for agriculture [8,9,10]. As an example, the improvement of biomass crops—which are all the plant species that can produce biomass to sustain biobased value chains [11,12]—could greatly benefit from this prospect. In fact, they include several under-domesticated species (see Mehmood, Ibrahim, Rashid, Nawaz, Ali, Hussain and Gull [12], and Pancaldi and Trindade [6] for a comprehensive list), whose breeding cycles are highly time-consuming [13], while several complex plant traits should be improved in these species to allow their cultivation on marginal lands to avoid competition with food production [6]. Nonetheless, the use of SQTLs in breeding contexts requires the availability of intraspecific allelic variability with a potential impact on traits of interest for the SQTL regions themselves, as genetic variability is the prerequisite for selection. Moreover, SQTLs must be able to predict the localization of causative loci of the traits for which they were mapped, also in other plant populations or species than the ones used for SQTLs identification.

The two latter aspects are the focus of this study. On the one hand, the intraspecific allelic genomic variability of cell wall SQTLs was assessed in six angiosperm species—*Arabidopsis thaliana*, *Solanum lycopersicum*, *Brassica napus*, *Zea mays*, *Oryza sativa*, and *Brachypodium distachyon*—for which multiple genomes representing diverse plant accessions were available. On the other hand, the SQTL value for predicting relevant loci in novel populations/species was tested by assessing the colocalization between cell wall SQTLs from *Miscanthus sinensis* and *Panicum virgatum* and genomic regions associated with cell wall variability identified through association mapping in miscanthus and switchgrass, respectively. Miscanthus and switchgrass are C4 biomass crops with high potential to fulfil diverse industrial applications, but at the same time still largely under-domesticated [5,13]. The study of the genetics underlying cell wall composition in these crops is thus pivotal to bringing them out of their current state [6,14]. The results obtained in this study demonstrate the validity of SQTLs for this purpose.

## 2. Results

### 2.1. General SQTL Alignment Patterns across Multiple Accessions of Six Angiosperm Species

To investigate the intraspecific genetic variability of SQTLs, the nucleotide sequences of 1184 cell wall SQTLs previously identified in six angiosperm species (Arabidopsis, rapeseed, tomato, maize, rice, and Brachypodium) [1] were aligned against 111 genome assemblies representing different plant accessions of the six species themselves (Figure 1). On average, alignments covered 89.3% of SQTLs initial bp length across all the genome assemblies tested (CV = 5.8%; Figure 2). Figure 2 shows that inter-specific syntenic regions spanned by SQTLs are, as expected, very well conserved overall, also at the intraspecific level. Nevertheless, it also highlights that minor SQTL portions display presence/absence variation (PAV) patterns between reference and specific target assemblies. Specifically, from the boxplots of Figure 2, these patterns appear particularly relevant in the three Poaceae species and, to a minor extent, tomato, where several SQTLs have relatively large portions of their total length not represented in alignments. On the one hand, this might be due to technological reasons such as the sequencing methodology of the target assemblies (e.g., de novo vs. reference-based resequencing). On the other hand, it could also depend on local intraspecific genomic rearrangements involving SQTLs regions. This hypothesis was tested in maize, by using the annotation of the transposon regions from the reference maize genome assembly used for initial SQTL detection (B73 version 4.0). This way, it was found out that maize SQTL regions span a large number of transposons (74 transposons per SQTL on average), which may be involved in intraspecific structural genomic variability, leading to the alignment patterns found (Appendix A).

Irrespective of the source of alignment length variability, the fact that SQTLs sometimes do not entirely align to target genome assemblies might lead to PAV of SQTL genes across the assemblies tested. In this sense, a BLAST validation of the SQTL genes displaying PAV according to the outputs of the intraspecific SQTL alignments revealed that 4225 SQTL genes are missing in one or more target assemblies across all the species tested (3.3% of all the genes contained in SQTLs). The majority of these genes is absent in only one target assembly (2086 genes across all the species), while 475 genes across all the species tested are missing in >50% of the target assemblies assessed for a particular species (Appendix A). Finally, of all the genes displaying PAV patterns between reference and target assemblies, 178 are cell wall genes (4.2%; Appendix A). Among these, some appear relevant for cell wall quality variability. These include an endoglucanase-coding gene from maize homolog to arabidopsis KORRIGAN (ND_NP_001288520.1, which is absent in four maize target assemblies), five PAL genes from *Brachypodium distachyon* involved in lignin synthesis (BG_XP_003575404.1, BG_XP_003575365.1, BG_XP_003575403.1, BG_XP_003575240.2, and BG_XP_003575238.1, which are absent in 11, 4, 2, 1, and 1 assemblies, respectively), and two CAD genes operating the in muro monolignol polymerization from *Brassica napus* (BL_XP_013702441.1, BL_XP_013702446.1, both absent in one target assembly) (Appendix A). To conclude, in spite of the genes just mentioned, the overall relatively low level of intraspecific PAV agrees with both the close relatedness of intraspecific accessions and the high level of syntenic conservation of SQTL regions.

### 2.2. Large Nucleotide Variation within the Intra-Specific SQTL Aligning Regions

SQTL alignments were also used to quantify intraspecific nucleotide variation at SQTL regions consisting of single nucleotide polymorphisms (SNPs) and insertions/deletions (INDELs, <100 bp) between reference and target assemblies (Figure 1). The results revealed extensive intraspecific nucleotide variability of SQTL regions (Figure 3). Specifically, SQTLs displayed on average 2.2 SNPs/kbp and 0.3 INDELs/kbp across all the species and assemblies assessed. These figures correspond to an absolute mean of 2643 SNPs and 295 INDELs per SQTL, considering the average SQTL length across all species (1176 kbp).

A more detailed analysis revealed that the number of SNPs/kbp and INDELs/kbp varies substantially between species (ANOVA’s *p* = 0.000; Figure 3A). Specifically, Arabidopsis and Brachypodium SQTLs displayed a particularly high number of SNPs/kbp between reference and target assemblies compared to the other species (LSD’s *p* < 0.001; Figure 3A). Conversely, regarding INDELs/kbp, only arabidopsis SQTLs displayed a substantially higher number of INDELs compared to the average across all SQTLs from all species (LSD’s *p* < 0.001; Figure 3A). Finally, both SNPs and INDELs occur with much higher frequency in dicot SQTLs (2.6 SNPs/kbp and 1.3 INDELs/kbp) compared to Poaceae SQTLs (0.8 SNPs/kbp and 0.2 INDELs/kbp; *t*-test’s *p* < 0.001 for both; Figure 3B), in spite of the opposite trend observed for overall coverage of SQTL alignments (Section 2.1).

The occurrence of SNPs and INDELs on SQTLs between reference and target assemblies was also assessed with respect to SQTL portions specifically spanned by genes. In this regard, it was found out that the majority of nucleotide polymorphisms occurs within intergenic genomic areas, as only 20.3% of SQTL SNPs and 11.5% of SQTL INDELs was located on SQTL genes (Appendix A). Moreover, the majority of all polymorphisms occurring on gene regions is located on noncoding gene segments, as only 44.4% of the SNPs on gene regions and 36.7% of the INDELs on gene regions were specifically located on exons. Still, in absolute terms, the latter two percentages correspond to an average of 159 SNPs and 23 INDELs per SQTL located on gene exons across all target species and assemblies, which are considerable numbers for the potential trait effects that these polymorphisms may cause. Finally, when assessed on a single-gene level, our results showed that 60,207 SQTL genes displayed at least one polymorphism in one species and against one target assembly (47% of all QTL genes). Of these, 3156 are cell wall-related genes (5.2% of total polymorphic genes). As average, each of these genes retains 16 SNPs and 3 INDELs across all species and all target assemblies, of which 7 SNPs and 1 INDEL are on exon regions.

### 2.3. Intraspecific SQTL Variability Leads to Changes in Cell Wall Protein Sequences with a Potential Functional Impact

The final step of the analysis of SQTL variability across the six species and related target genome assemblies consisted in assessing the effect of SNPs and INDELs on translated protein sequences. Overall, across all SQTLs, all species, and all target assemblies, 30,654 SQTL genes (23% of all SQTL genes) displayed polymorphisms leading to one or more protein sequence changes between reference and one or more target assemblies (Appendix A). Of these, 1861 are cell wall genes. On average, each of the SQTL genes displaying protein-impacting polymorphisms across target assemblies retained 5.1 SNPs and 0.2 INDELs with an effect on the translated protein sequences. These polymorphisms cause a mean of 4.1 amino acid changes per coded protein sequences. Finally, 1126 SQTL genes of all the ones displaying protein-impacting polymorphisms (0.8% of all SQTL genes) have SNPs or INDELs between reference and target assemblies that lead to stop codons and (likely) truncated proteins. Of these, 421 are cell wall genes.

To assess the potential effect of the intraspecific protein sequence variability on cell wall quality traits, how such patterns impact functionally relevant and highly syntenically conserved candidate SQTL genes identified in our previous SQTL study was studied [1]. These genes include members of the *COBRA* (*COB*) and *COBRA*-like (*COBL*) gene families responsible of the Brittle-culm cell wall rice mutants [15]; important genes associated with lignin content variability in cell walls, as *FERULATE 5-HYDROXYLASE* (*F5H*), *CINNAMOYL-CoA O-METHYLESTERASE* (*CCOAOMT*), and *PHENYLALANINE AMMONIA-LYASE* (*PAL*) [16]; key hemicellulose- and cellulose-related genes for cell wall polysaccharides metabolism, as members of the *CELLULOSE SYNTHASE* (*CESA*), *CELLULOSE SYNTHASE-LIKE* (*CSL*), and *IRREGULAR XYLEM* (*IRX*) gene families [17,18]; important transcription factors for secondary cell wall development such as *WRKY12*, *NAC SECONDARY WALL-THICKENING FACTOR1* (*NST1*), and *C3H14* [19,20]. For all of these genes, protein sequences were aligned and annotated for functional domains and motifs, allowing for the detection of protein sequence variability across assemblies, as well as for the assessment of the effect of variability on protein polarity, hydrophobicity, and functional domain properties. Figure 4 and Figure 5 and Appendix A display the results of these analyses.

Among all the alignments computed, the ones of the three members of the rice *BRITTLE CULM-LIKE* loci *OsBCL1*, *OsBCL8*, and *OsBCL9*, and of their related syntenic homologs conserved through SQTLs [1], yielded particularly interesting results. In fact, all the proteins coded by these three rice genes displayed one or more amino acid changes with effect on the polarity, charge and/or hydrophobicity of the proteins themselves. Specifically, these changes took place in the rice cultivars “Carolina Gold Select” and “Azucena” compared to the reference rice “Nipponbare”. In this regard, *OsBCL9* displayed three amino acid changes within the *COBRA* domain of the protein, of which one involving a change in polarity (alanine to threonine at position 123) and leading to the removal of a β-sheet domain in the 2D protein configuration (Figure 4A). A similar pattern was observed for *OsBCL1* (Appendix A). Finally, *OsBCL8* displayed a cysteine to arginine change, leading to loss of amino acid charge within the protein C-terminus (position 653), right next to the GPI-anchoring-related ω-site (Figure 4B). In addition to the rice *BCL* loci, the other *BCL* genes from Brachypodium, sorghum, and maize syntenic to *OsBCL* genes and retained within SQTLs were also assessed for protein variability. Intraspecific alignments revealed that the Brachypodium homolog of the *OsBCL8* gene (XP_003559754.1) displayed massive protein rearrangements in three of the 45 Brachypodium genome assemblies analyzed (“ABR3”, “Bd2”, and “Adi12”). These changes involve large deletions of the *BdBCL8* protein sequence, including a consistent part of the *COBRA* domain in the line “ABR3”, and the complete *COBRA* domain in the lines “Bd2” and “Adi12” (Figure 5).

In addition to the particularly relevant examples found for the *BRITTLE CULM*-like genes, intraspecific amino acid changes and INDELs impacting protein properties were found in several of the other proteins coded by the candidate SQTL genes inspected (Appendix A). As for the rice brittle culm loci, protein changes were observed both within and outside protein functional domains or motifs. For example, a maize *IRX9* protein involved in xylan biosynthesis (NP_001147664.1) displayed two regions with relatively large INDELs and different amino acid substitutions within the GT43 functional domain (Appendix A). These changes take place in 12 of the 33 maize assemblies assessed and have effects on protein polarity and/or charge. Similarly, the enzyme coded by the Brachypodium secondary cell wall *CESA7* gene (XP_003574029.1), also retained within SQTLs, displayed large deletions within the cellulose synthase catalytic domain in the accession “TR9” compared to the reference Brachypodium genome (Appendix A). Moreover, amino acid substitutions within the protein functional domain with an effect on protein polarity were also observed. Overall, these patterns were linked to major changes in the secondary structure of the *CESA7* protein, including the deletion of seven α-helix and one β-sheet domains (Appendix A). Finally, multiple INDELs and amino acid substitutions with impact on protein chemical properties were also detected across the arabidopsis accessions for the *AtC3H14* gene (a transcription factor regulating secondary cell wall thickening), across different Brachypodium accessions for the *BdWRKY12* gene (a major transcription factor regulating cell wall biosynthesis in plant vessels), and between the rice reference genome and three rice cultivars (Koshihikari, Kitaake, and Azucena) for the *OsNAC43* gene (also a major transcription factor for secondary cell wall biosynthesis in different plant tissues) (Appendix A).

### 2.4. SQTLs Are Valid Tools to Predict Important Cell Wall Genomic Loci in Miscanthus sinensis and Panicum virgatum

In addition to evaluating the level and relevance of the intraspecific genetic variability of SQTLs in diverse species, another aim of this study was to assess the validity of SQTLs for predicting genomic loci associated with biomass quality traits in novel plant populations and species. This aspect was studied by assessing the degree of colocalization between the SQTLs previously detected in *Miscanthus sinensis* and *Panicum virgatum* [1] and cell wall related QTLs mapped by genome-wide association analysis (GWAS) in a *Miscanthus sinensis* collection and QTLs mapped by other researchers in an F1 progeny of a biparental cross of two switchgrass lines diverging for cell wall quality traits [21]. These populations represent respectively a different intraspecific population than the one from which QTLs used for SQTLs mapping came from (miscanthus), and a separate species than the ones from which initial QTLs have been selected at the moment of SQTLs detection (thus, a hypothetical novel, under-domesticated, species; switchgrass).

The GWAS conducted on the miscanthus population identified a total of 91 QTLs associated to eight traits related to cell wall content and composition (Figure 6 and Appendix A). These 91 QTLs cover 6.8% of the miscanthus genome and include a total of 148 cell wall genes. First, the general degree of overlap between the GWAS QTL regions and the 254 SQTLs previously detected in miscanthus (which cover 32.7% of the miscanthus genome; Appendix A) was assessed. This analysis revealed that 67 SQTLs (26.4% of all the miscanthus SQTLs) colocalized (for parts of their regions) with the 91 GWAS QTLs. Moreover, it was observed that 35 of the 91 GWAS QTLs (38%) colocalized for >50% of their bp length with genomic regions where miscanthus SQTLs are also present (Figure 6). To test if these figures highlight significant colocalization of the QTL loci identified by GWAS with the miscanthus SQTLs, a permutation analysis was performed by constructing 100 sets of 91 random genomic regions mirroring the bp size distribution of the GWAS QTLs (Appendix A). For each set, the proportion of random QTLs colocalizing with SQTLs was computed, and a binomial test was performed to assess if this proportion was significantly lower than the one observed for the real GWAS QTLs. The results showed that, as an average across the 100 random QTLs sets analyzed, 17 QTLs of the 91 included in each set (19%) colocalized for >50% of their length with SQTLs. This figure is significantly lower compared to what observed in real QTLs (binomial test significant in 91 of the 100 tests performed at α = 0.01), highlighting that SQTLs colocalize significantly with the GWAS QTLs.

An analogous procedure to the one just described for miscanthus was performed in switchgrass to test the colocalization between the 56 cell wall-related QTLs identified by Ali, Serba, Walker, Jenkins, Schmutz, Bhamidimarri, and Saha [21] (Appendix A), and the 254 SQTLs previously detected in switchgrass and conserved across Poaceae (Appendix A). In switchgrass, it was shown that 53 SQTLs (20.8% of all the switchgrass SQTLs) (partially) colocalize with the 56 cell wall QTLs, while 33 QTLs (59%) colocalize for >50% of their length with SQTLs (Figure 7). Permutation analysis demonstrated that the colocalization of QTLs and SQTLs is significant, as random QTL sets produced significantly lower proportions of colocalizing QTLs (binomial test significant in 99% of iterations at α = 0.01) (Appendix A).

Since colocalization between SQTLs and QTLs was demonstrated for both miscanthus and switchgrass, as a final step, we analyzed which cell wall genes are retained in co-localizing regions between SQTLs and QTLs, as well as the level of syntenic conservation of those genes in angiosperms. In this regard, miscanthus results showed that several important cell wall genes were retained within the colocalizing regions between SQTLs and GWAS QTLs. These include, among others, different central cell wall transcription factors, as one of the two *MsBLH6* (GQ_01G471400 and GQ_02G130600), the *MsWRKY12* (GQ_12G168300), the miscanthus homolog of arabidopsis *VND4* (GQ_12G158800), and the *MsMYB103* (GQ_07G174800) (Figure 6 and Table 1). In addition, they comprise important lignin genes, such as a miscanthus *PAL* and *CCOAOMT* copy (GQ_12G150500 and GQ_07G169800), or the miscanthus homolog of the maize *BRITTLE-STALK 2* locus (*COBL* gene; GQ_12G165800) (Figure 6 and Table 1). Finally, synteny analysis showed that each of these genes is syntenic to other 84 genes on average (range 26–190). Most of the synteny is, as expected, toward Poaceae (Figure 8A,B). However, some syntenic connections involve also relatively distant eudicot species, such as rapeseed or tomato, which have copies of their *WRKY12*, *VND4*, and *CCOAOMT* genes syntenic to miscanthus.

For switchgrass, the most relevant cell wall genes found in colocalizing regions between SQTLs and QTLs from Ali, Serba, Walker, Jenkins, Schmutz, Bhamidimarri, and Saha [21] are summarized in Figure 7 and Table 2. Among others, these include a homolog of the arabidopsis *FRAGILE FIBRE 1* locus (*FRA1*; IV_6NG323500) and a homolog of the arabidopsis *KOR* gene (IV_1NG408000). Moreover, an *IRX* gene involved in xylan synthesis (IV_5NG144100), and two *MYB42*/*MYB85* genes important for lignin biosynthesis (IV_2NG449200 and IV_6NG352900). All these genes are retained in QTLs from Ali, Serba, Walker, Jenkins, Schmutz, Bhamidimarri, and Saha [21] that are associated to traits for which the genes themselves appear highly functionally relevant (Table 2). Moreover, synteny analysis of these genes showed that, as for miscanthus, they are highly syntenic, even if synteny is again mostly restricted to Poaceae (Figure 8C,D).

### 2.5. SQTLs as Tools to Circumvent Limitations of Genetic Mapping Approaches

In the final step of this research, it was evaluated whether SQTLs, given the incorporation of information from multiple genomic loci previously shown to determine variability in a trait of interest (in this case cell wall quality), can be used to overcome limitations of genetic mapping approaches. Specifically, a common issue encountered in association mapping—and especially GWAS—is that if patterns of population structure overlap with patterns of phenotypic variability across accessions for the trait(s) evaluated, population structure correction can hide (part of) the relevant loci that govern the trait(s) [36]. The GWAS conducted in *Miscanthus sinensis* was therefore used to study if SQTLs can, at least partly, overcome this limitation. First, the analysis of population structure computed to perform the “standard” GWAS described in Section 2.4 was co-analyzed with the phenotypic data on cell wall traits to assess covariation between population structure and phenotypic traits across the miscanthus accessions of the GWAS panel (Appendix A). Interestingly, this covariation was found for multiple traits tested, including total cell wall (NDF; dry matter percentage), total ADF (dry matter percentage), cellulose (dry matter percentage), cellulose (NDF percentage), hemicellulose (dry matter percentage), and hemicellulose (NDF percentage; Appendix A). A GWAS without population structure correction was therefore run, and colocalization between SQTLs and the LOD peaks found by this GWAS and not already included among the 91 QTLs from the “standard” GWAS with population structure correction (Section 2.4) was assessed. These peaks would normally be discarded as false positive associations due to population structure. However, their colocalization with SQTLs indicates that in other species, QTLs for similar traits were found on these exact genomic regions, highlighting their potential relevance. As an extra control, the SNP markers included within the LOD peaks colocalizing with SQTLs and not included among the 91 QTLs of the “standard” GWAS were extracted and tested with ANOVA for significant differences in the major allele frequency (MAF) between the different population structure groups depicted in Appendix A. Upon detection of significant MAF differences between covarying population structure and phenotypic groups, the peak regions were declared as “extra QTLs” found by colocalization with SQTLs (Figure 9).

In total, 17 “extra QTLs” were detected (Appendix A). Their analysis demonstrated that they contain a total of 13 cell wall genes. Among these, the most relevant ones appeared to be a miscanthus homolog of the arabidopsis *KOR* gene (GQ_05G142200) that was found within a QTL associated to cellulose dry matter content. Moreover, an *MsCSL* gene (GQ_19G142500), a homolog of the Arabidopsis and maize *CslD5*, was also found within four colocalizing “extra QTLs” associated with total cell wall dry matter content, ADF dry matter content, cellulose dry matter content, and cellulose NDF percentage. Finally, two peroxidases (*PRX*; GQ_06G157300, GQ_19G143200), and a *XYLOGLUCAN ENDO-TRANS-GLYCOSYLASE/HYDROLASE* (*XTH*; GQ_19G142300).

## 3. Discussion

### 3.1. SQTLs as Reservoirs of Allelic Variation with a Potential Use for Biomass Improvement

In a previous publication, it was shown that SQTLs are useful breeding tools to project known genetic information on the architecture of traits of interest—specifically cell wall quality traits—from model species to understudied crops [1]. This research aimed at assessing whether SQTLs can potentially guide breeding activities by spanning genomic regions displaying allelic variability for target traits and by pinpointing relevant loci associated with biomass quality variability in novel plant populations and crops. Regarding the first point, this study clearly showed that SQTL regions are reservoirs of intraspecific allelic variability. A minor part of this variability entails PAV of SQTL genes. Gene PAV is commonly found in intraspecific comparative genomic studies, such as pangenome analyses, and is a known source of trait variability [37,38]. In this sense, the SQTL cell wall genes displaying intraspecific PAV may potentially impact biomass traits in target accessions in several ways. For example, in the case that a SQTL gene is part of a multigene family, its PAV could affect overall gene copy number and lead to differential gene dosage across accessions [39,40]. Alternatively, in more extreme cases, PAV may lead to a loss-of-function, with a likely large impact on plant traits [40]. In this study, the SQTL genes showing PAV between reference and target assemblies included *PAL* and *CAD* genes, which displayed PAV among some of the *Brachypodium distachyon* and *Brassica napus* accessions assessed. *PAL* and *CAD* are multigene families, and gene dosage is thought to be important for their functionality [41,42,43]. Moreover, in the close relative of *Brassica napus*, *Brassica rapa*, intraspecific copy number and PAV of *PRX* genes, which together with *CAD* determine the efficiency of lignin production at the final steps of the lignin pathway [44], was suggested to affect lignin metabolism and related morphological traits [45]. Therefore, intraspecific PAV of SQTL genes might be relevant for determining variability in cell wall quality traits, and genomic analysis of SQTL regions in (novel) crop panels might quickly highlight promising target accessions for inclusion in breeding programs based on the assessment of gene PAV.

In addition to PAV, nucleotide polymorphisms were also found in high numbers in the intraspecific analyses of SQTL regions. Specifically, multiple genes that were previously defined as relevant SQTLs candidates [1] display intraspecific nucleotide polymorphisms that impact, sometimes considerably, the sequences of their proteins. In this sense, the examples reported for the rice *BCL8* locus and its Brachypodium syntenic homolog identified through SQTLs are particularly relevant. In fact, the protein modifications reported for these genes sometimes break the integrity of the *COBRA* domain of the protein or affect the membrane-anchoring domains at the C-terminus. These mutations are highly similar to the ones found in the well characterized *OsBC1* and *OsBCL1* mutants, which are close homologs of the *OsBCL8* gene [46,47]. Both these mutants display alterations of the plant mechanical strength and of cell wall compositional properties as effect of the *COBL* gene mutations [15,46,47]. Thus, the mutations found in this study might potentially lead to similar effects. Interestingly, the rice cv. “Azucena”—which is one of the rice cultivars displaying a change in protein polarity in the C-terminus of the *BCL8* protein—displays differences in the relative content of cell wall components, including cellulose, compared to the reference of this study, cv. “Nipponbare” [48]. Nevertheless, future research is needed to determine a relationship between these mutations and cell wall composition. The same goes for all the other mutations discussed in Section 2.3.

To conclude, irrespective of the functional relevance of the specific patterns of genomic variability observed for the SQTL regions in the intraspecific comparisons performed, it is still highly relevant that SQTL genes were proven to show such patterns. This demonstrates that the high conservation of gene presence and order across diverse genomes does not preclude the existence of intraspecific allelic variability for those highly conserved regions. Therefore, synteny and SQTLs can be effectively used to mine interesting alleles by genomically analyzing SQTL regions in target species and searching for specific mutations previously identified as particularly relevant in model crops. If this step would be performed during the screening of useful material for breeding programs, it would lead to faster and more effective breeding activities in novel (under-domesticated) biomass species.

### 3.2. SQTLs as Valid Tools toward the Improvement of Breeding Activities in Novel Crops

In the second part of this study, it was tested if SQTLs represent effective tools to predict relevant loci associated to biomass quality in novel mapping panels or crop species. The results showed that two sets of QTLs from *Miscanthus sinensis* and *Panicum virgatum*, respectively, colocalized significantly with SQTL regions of these two species as compared to random genomic loci. This result demonstrates that, in a breeding context involving novel species, SQTLs can be valuable tools to pinpoint relevant target loci that are likely responsible of variability in traits of interest. Moreover, by using the annotations of SQTL genes from the multiple species that are represented within SQTLs—as performed here for cell wall genes—it is possible to filter candidate genes based on functional and literature information. Finally, syntenic conservation of target loci through SQTLs can quickly display the functional conservation of interesting target genes through multiple (model) species for which studies might be available to better evaluate the functional relevance of cell wall genes within QTLs and SQTLs.

By applying the approach just described, in addition to the colocalization between SQTLs and miscanthus and switchgrass QTLs, several genes involved in these colocalizations were identified that represent valuable candidates for determining variability in biomass composition in these species. Specifically, some of the genes found from the miscanthus GWAS and conserved through SQTLs appear of particular interest. For example, the transcription factor *BLH6* is known to deeply affect the properties of plant cell walls, not only within different species [23,49], but also between different species clades, such as Poaceae vs. eudicots [20]. Therefore, the finding of this gene within GWAS QTLs colocalizing with SQTLs makes it a very good candidate for modulating the relative ratio of different cell wall polysaccharides and lignin in miscanthus. In addition to *BLH6*, genes such as *WRKY12*, *VND4*, and *MYB103* are all central cell wall genes, demonstrating the modulation of cell wall regulation and composition across a range of species [19,20,28,50,51,52,53]. All these genes are thus important candidates for the cell wall quality variability observed in the miscanthus panel used for the GWAS described in this manuscript. Ideally, this information might be therefore used to map relevant mutations at these genes and screen plant material for those mutations. Remarkably, the intraspecific genomic comparisons performed in this research revealed that nucleotide variability leading to amino acid substitutions and protein INDELs were observed among the Brachypodium accessions. These two approaches—SQTL-guided association mapping and genomic screening/prediction of mutations at target candidate genes—could be combined in novel species to enable quicker (pre-)breeding for biomass compositional traits.

In the last part of this study, whether SQTLs can be used to make the population structure correction more efficient in GWA studies was also tested. While it needs to be clearly stated that the accounting of population structure in GWAS is pivotal to enabling accurate mapping, the results obtained here indicate that the inspection of colocalization between SQTLs and significant SNPs normally discarded as false-positives after population structure correction could represent a useful complement of a “standard” GWAS pipeline. Specifically, as they are defined, SQTLs represent genomic regions that are syntenically conserved between species and in multiple model species were shown to be associated to QTLs for a trait of interest [1]. Therefore, they could help in retrieving some of the “missing heritability” that can get blurred below the threshold of false discovery [36], in the case of the SQTL areas significantly covarying with a target trait in other (related) crops. However, the “eye” of the researcher is crucial in order to be able to evaluate the relevance of potential candidate genes found within the “extra GWAS QTLs” pinpointed by using SQTLs. In this sense, in our miscanthus panel, some of the candidate genes retained within the 17 detected “extra QTLs” of Section 2.5 appear highly relevant in the context of cell wall variability. Therefore, these genes might be considered, depending on the needs, in breeding settings to improve miscanthus biomass quality, together with all the other ones discussed in the previous paragraph(s).

## 4. Materials and Methods

### 4.1. Intraspecific SQTL Alignment

Multiple chromosome-level genome assemblies representing different accessions of *Arabidopsis thaliana*, *Brassica napus*, *Solanum lycopersicon*, *Brachypodium distachyon*, *Oryza sativa*, and *Zea mays* were collected from online databases after the literature search (Appendix A). These six species were chosen because of the availability of multiple good-quality chromosome-level genome assemblies from either pan-genomic studies [37,54,55] or genomic databases, as well as their relevance for plant research. Moreover, they include both eudicots and grasses for which SQTLs were available from our previous study [1]. The reference assembly of each of the six species above which SQTLs were initially detected was used to extract the reference nucleotide sequences of cell wall SQTLs. These sequences were then aligned against the accessions collected by using the NUCmer package of the MUMmer software [56,57]. NUCmer was run with the following parameters: --minmatch 100 and --mincluster 200. The NUCmer show-snps command was also called (default parameters) to detect SNPs and INDELs between reference SQTL sequences and aligning regions on target accessions.

### 4.2. Analysis of Intraspecific SQTL Alignments

Custom R scripts were developed to process the NUCmer outputs and extract different information. First, the aligning regions and coordinates of SQTLs in every target accession; second, SQTL coverage for each alignment; third, the SQTL (cell wall) genes included in each alignment; finally, the SNPs and INDELs found along alignments. To extract these data, the custom R scripts made also use of a previously developed list of cell wall genes from 169 angiosperm genomes [1], as well as of the IRanges and GenomicRanges R packages [58]. Moreover, to quantify gene PAV in SQTLs alignments, the data on missing genes in alignment produced by NUCmer were validated by a BLAST search [59] of the genes identified as missing in specific alignments against the assemblies involved in the alignments themselves (*E*-value = 1 × 10^−3^). Finally, all the statistical analyses reported in this manuscript were performed using R or SPSS v27.0 (IBM Corp., Armonk, NY, USA).

### 4.3. Analysis of Cell Wall Protein Sequence Changes

The assessment of the effects of SQTLs genomic variability on protein sequences was performed by using a custom R script were the coordinates of the exons of every SQTL gene included in alignments and concurring to code the main gene transcripts were used to retrieve gene CDS and translated proteins in the target genome assemblies. On the retrieved proteins, sequence changes between reference and target assemblies were assessed with ClustalW [60]. Moreover, HMMsearch [61] was used to annotate protein functional domains available on the PFAM database [62]. Furthermore, protein signal peptides, including N- and C-terminus and related functional signals were annotated by integrating the predictions provided by SignalP v6.0 [63], DeepTMHMM [64], NetGPI [65], and DeepLoc v2.0 [66]. Finally, changes in protein structure and properties due to sequence changes were assessed by using NetSurfP v3.0 [67].

### 4.4. Genome-Wide Association Study on Miscanthus sinensis

The GWAS on *Miscanthus sinensis* was performed by using a miscanthus collection established in 2013 in Wageningen (The Netherlands) and composed of 94 accessions originated from various international gene banks and breeding programs. Genotypes were planted in square-like plots with 16 clonal replicas, and the four central plants of each plot were harvested every spring for five years, starting in 2017.

Phenotyping was performed for eight cell wall quality traits, including NDF cell wall as percentage of dry matter, ADF cell wall as percentage of dry matter, ADL lignin as percentage of dry matter, lignin as percentage of NDF, cellulose as percentage of dry matter, cellulose as percentage of NDF, hemicellulose as percentage of dry matter, and hemicellulose as percentage of NDF (Appendix A). Phenotyping was performed by first chopping the harvested miscanthus stems into pieces of 4 cm, and by drying (60 °C, 48 h) and weighing stem pieces to determine dry matter content. The dried stems were then milled, and a subset of samples was used for training a near-infrared spectrometry (NIRS) model for cell wall composition, by performing NDF, ADF, and ADL biochemical analyses following the ANKOM Technology protocols (ANKOM Technology Corporation). The NIRS model was in turn used to phenotype the cell wall traits on the milled feedstock of all the genotypes of the collection.

The genomic DNA of all the accessions was isolated from random young leaves from the four central plants of all the plots in the collection, following a CTAB-based protocol [68]. Extracted DNA from every sample was digested by using the restriction enzyme EcoR1, ligated to unique adapters, pooled, purified, amplified, and finally sequenced using an Illumina HiSeq X10/4000 system. Sequencing was performed by BGI (Shenzen, Guangdong, China), and generated 371.52 Gb of cleaned data. Reads were aligned to the *Miscanthus sinensis* reference genome [69], resulting in the identification of ~7.0 million SNPs. SNPs were filtered for only biallelic SNPs displaying 100% call rate across the 94 miscanthus accessions and a minor allele frequency > 20%. Moreover, following chromosome-wide LD analysis with the LD.decay function from R package Sommer [70] (Appendix A), SNPs were further filtered to keep only one marker for genomic bins corresponding to one third of the average chromosomal LD distance. This way, a set of 57,891 markers relatively evenly distributed over the 19 miscanthus chromosomes was obtained.

The filtered SNPs were used to estimate population structure by using van Raden kinship [71] and principal coordinate analysis (PCoA; ape R package—Paradis and Schliep [72]) (Appendix A). A dendrogram of the kinship matrix was also produced (ape R package—Paradis and Schliep [72]). Moreover, patterns of population structure among accessions were compared with the ones inferred from a principal component analysis (PCA) on the phenotypic data (Appendix A), to assess covariation between population and phenotypic accession clusters (Appendix A).

Genome-wide associations between the filtered SNPs and the eight cell wall traits above were performed by using a linear mixed model (LMM) incorporating SNP data and the kinship matrix, as implemented in the statgenGWAS R package [73]. FDR correction (1%) was used to account for multiple testing [74], while QQ-plots of observed vs. expected p-values of associations were computed to assess the effectiveness of population structure correction (Appendix A). GWAS analyses were performed separately for each trait. Chromosome-scale LD windows were used to define significant regions around the significant markers found, within which candidate genes were looked for, by using the set of angiosperm cell wall genes developed in our previous SQTL study [1] and the Arabidopsis- and rice-based annotations of the miscanthus genes from Phytozome. Moreover, a separate GWAS was also performed without incorporating the kinship matrix in the model, to perform the study described in Section 2.5.

### 4.5. Colocalization of SQTLs and Cell Wall Loci Mapped on M. sinensis

Colocalization between SQTLs and the 91 QTLs found by the miscanthus GWAS was performed by developing 100 sets of 91 random QTL regions from the miscanthus genome mirroring the size distribution of the QTLs from GWAS results (custom R script). The proportion of QTLs colocalizing for >50% of their bp length with SQTLs was then calculated for every set, and binomial tests were performed to assess if random QTLs co-localized with SQTLs significantly less than the QTLs from the GWAS (custom R script).

In addition to calculating the statistical significance of the colocalization between miscanthus SQTLs and QTLs, the cell wall genes in colocalizing regions were identified by using the set of angiosperm cell wall genes developed in our previous SQTL study [1]. Moreover, angiosperm-wide syntenic conservation of those genes was analyzed by retrieving their syntenic homologs from the synteny network developed by Pancaldi, et al. [75] and Pancaldi, Vlegels, Rijken, van Loo, and Trindade [1].

### 4.6. Retrieval of the P. virgatum Cell Wall QTLs and Analysis of Their Colocalization with SQTLs

A total of 56 QTLs related to cell wall traits in switchgrass were retrieved from the results of Ali, Serba, Walker, Jenkins, Schmutz, Bhamidimarri, and Saha [21] (Appendix A). Colocalization between these QTLs and the switchgrass SQTLs was analyzed with an analogous procedure to miscanthus. Specifically, 100 sets of 56 random QTL regions from the switchgrass genome mirroring the size distribution of the 56 QTLs from Ali, Serba, Walker, Jenkins, Schmutz, Bhamidimarri, and Saha [21] were computed. The proportion of QTLs colocalizing for >50% of their bp length with SQTLs was then calculated for every set, and binomial tests were performed to assess presence and significance of a decrement in such proportion (custom R script).

As performed in miscanthus, the cell wall genes in colocalizing regions were identified by using the set of angiosperm cell wall genes developed in our previous SQTL study [1]. Moreover, angiosperm-wide syntenic conservation of those genes was analyzed by retrieving their syntenic homologs from the synteny network developed by Pancaldi, van Loo, Schranz, and Trindade [75] and Pancaldi, Vlegels, Rijken, van Loo, and Trindade [1].

## 5. Conclusions

This study aimed at assessing the validity of SQTLs to assist breeding of (novel) biomass crops, by (i) analyzing the intraspecific level of SQTLs allelic genomic variability in panels of available good-quality plant genomes, (ii) assessing the SQTLs predictive value of relevant biomass-quality-related loci in miscanthus and switchgrass, and (iii) suggesting possible side-uses of SQTLs to complement standard approaches of genetic mapping. The results showed that SQTLs are valuable tools to improve breeding activities in novel crops, where they can be applied in different ways to either screen plant material for genes or alleles of interest, or to pinpoint relevant loci based on the information from model species in novel accession panels or crops. Moreover, the research performed to achieve the goals above allowed the finding of relevant alleles at candidate SQTL loci, as well as of important candidate cell wall genes for miscanthus biomass improvement. Future research could investigate the phenotypic relevance hypothesized for the intraspecific patterns of SQTLs variability revealed in this study, by phenotyping cell wall composition in the plant lines showing promising variability. Moreover, the miscanthus loci highlighted in this study could be targeted by genetic modification to study their specific relevance for miscanthus biomass quality. Finally, SQTLs might start to be included in breeding programs of under-domesticated biomass crops as proposed in this research, to test their usefulness in novel, important, breeding settings.

## Figures and Tables

**Figure 1 plants-12-00779-f001:**
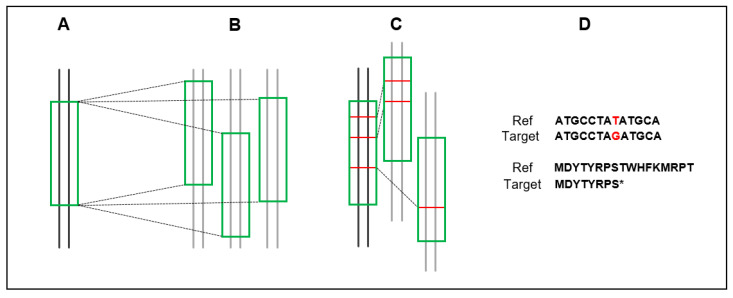
Schematic representation of the workflow followed to analyze intraspecific allelic variability of SQTLs. SQTL nucleotide sequences have been extracted from reference genome assemblies of six species (**A**) and aligned against multiple genome assemblies representing diverse accessions of each species (**B**). Nucleotide polymorphisms and gene presence–absence variation were quantified and analyzed across accessions and SQTLs (**C**). Finally, the effect of genomic polymorphisms on gene coding sequences on protein sequences and structures was also assessed (**D**) and compared with known mutations responsible of relevant biomass phenotypes.

**Figure 2 plants-12-00779-f002:**
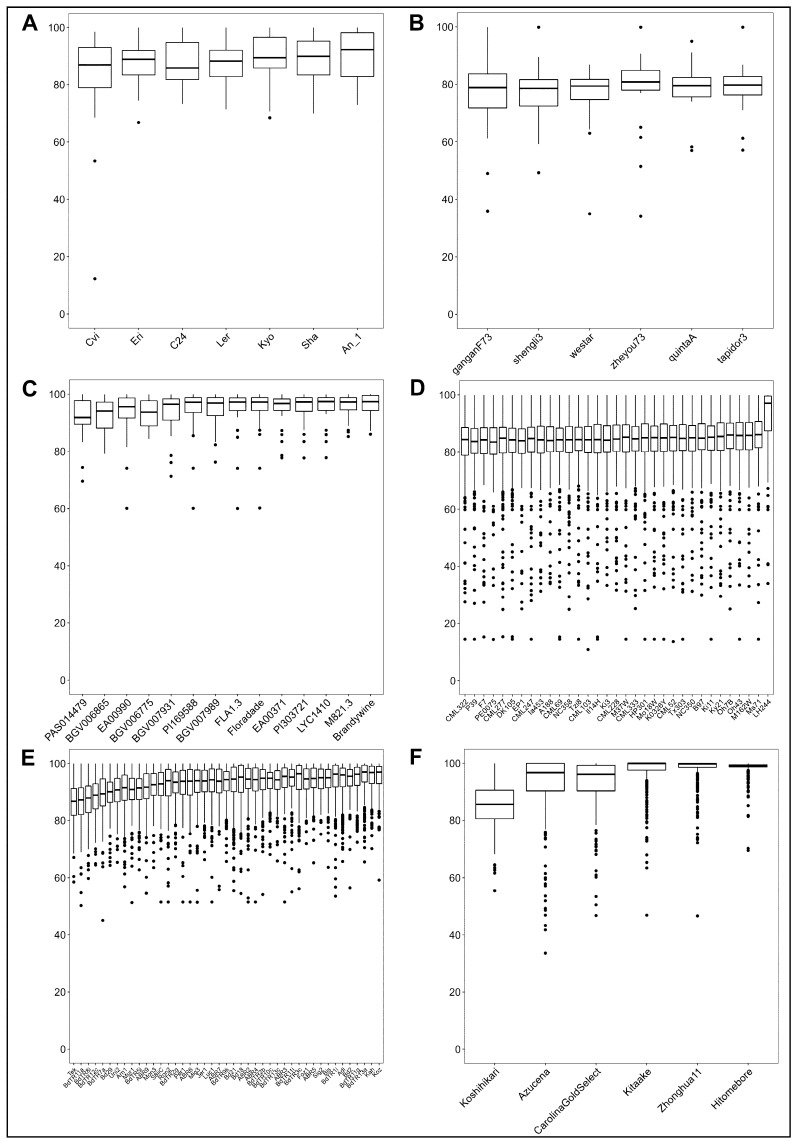
Boxplots representing the coverage of reference SQTL segments across the target accessions used for SQTL alignment in every species. Data points are percentages of SQTL segments contained in a target assembly (one data point per SQTL, per species). (**A**) *Arabidopsis thaliana*; (**B**) *Brassica napus*; (**C**) *Solanum lycopersicum*; (**D**) *Zea mays*; (**E**) *Brachypodium distachyon*; (**F**) *Oryza sativa*.

**Figure 3 plants-12-00779-f003:**
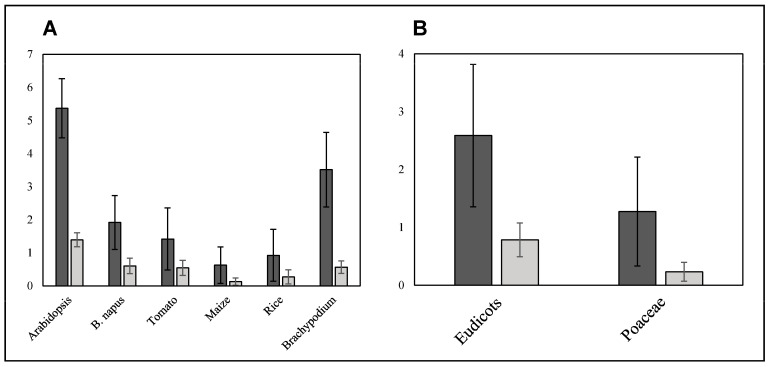
Average number of SNPs/kbp (dark grey) and INDELs/kbp (light grey) over SQTLs and per species (**A**) and plant clade (Eudicots vs. Poaceae) (**B**).

**Figure 4 plants-12-00779-f004:**
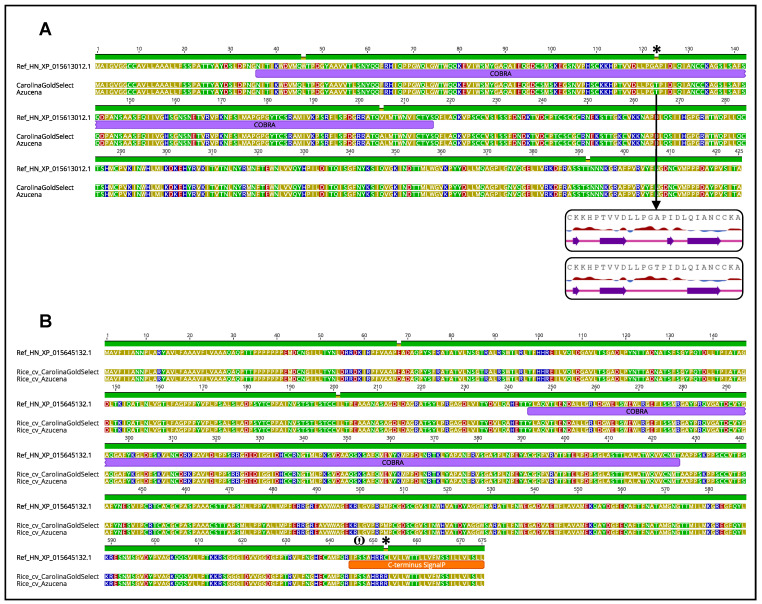
Amino acid changes and their effects on protein structure for three *brittle culm*-like loci between the reference rice cv. “Nipponbare” and the two cultivars “Azucena” and “CarolinaGoldSelect”. (**A**) *OsBCL9*; (**B**) *OsBCL8*. In each figure, protein sequences are colored according to amino acid polarity, and protein domains and signal peptides are annotated. Amino acid changes indicated with * indicate a change in polarity, while sites annotated with ω indicate the predicted GPI-anchoring-related omega sites.

**Figure 5 plants-12-00779-f005:**
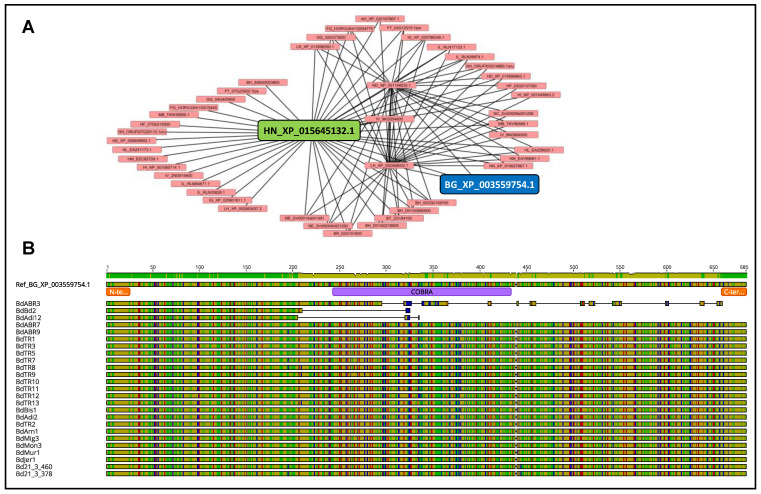
The syntenic conservation of the rice locus *OsBCL8* (HN_XP_015645132.1) across Poaceae through SQTLs (**A**), and the intraspecific allelic variability with an impact on protein sequence of the *Brachypodium distachyon* syntenic homolog of *OsBCL8* conserved through SQTLs (BG_XP_003559754.1), across 25 Brachypodium accessions compared to the reference genome (**B**). In Figure 5A, connections between genes indicate synteny through SQTLs, as described in [1]. In Figure 5B, protein sequences are colored according to amino acid polarity, and protein domains and signal peptides are annotated.

**Figure 6 plants-12-00779-f006:**
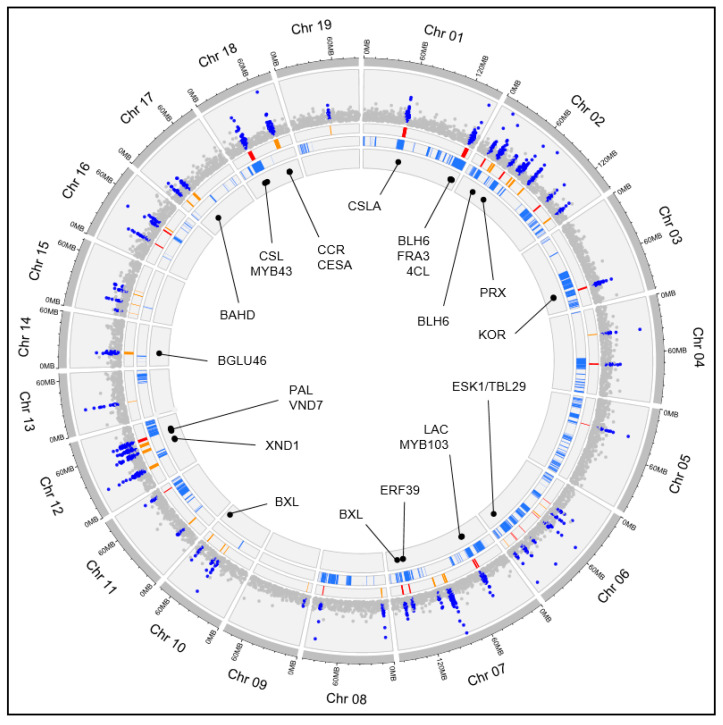
The results of the miscanthus GWAS and of the colocalization between miscanthus SQTLs and QTLs plotted onto the miscanthus genome. From outside to inside, the first strip shows the LOD scores of the markers from the GWAS, with the markers included in the 91 QTLs colored in blue; the second strip displays the genomic ranges of the 91 QTLs from the GWAS, with the QTLs colocalizing with SQTLs for >50% of their length colored in red; the third strip displays the positions of miscanthus SQTLs; the fourth strip highlights the most relevant candidate genes identified from the GWAS analysis.

**Figure 7 plants-12-00779-f007:**
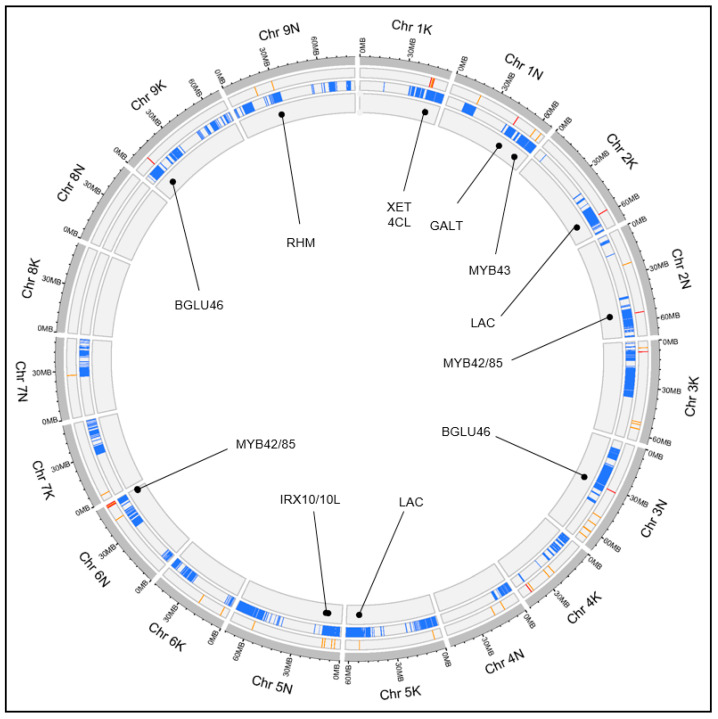
The genomic map of the cell wall QTLs collected for switchgrass from the work of [21] and of the switchgrass SQTLs. From the outside, the first band represents the position of the 56 switchgrass QTLs collected from [21], with the QTLs colocalizing with SQTLs colored in red. The second band represents the positions of the switchgrass SQTLs. The third band displays relevant cell wall candidate genes found in the colocalizing regions between SQTLs and QTLs.

**Figure 8 plants-12-00779-f008:**
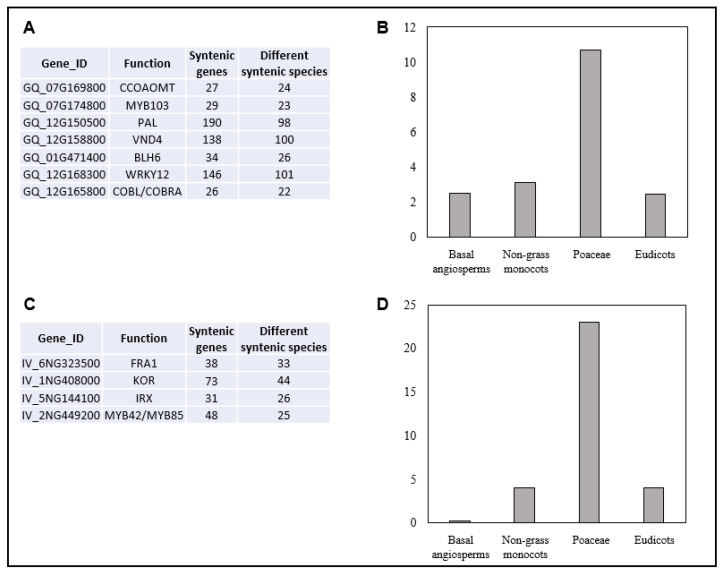
Analysis of the syntenic relationships of the most relevant genes from miscanthus (**A**,**B**) and switchgrass (**C**,**D**) that are conserved through SQTLs and colocalizing with the QTLs described in Section 2.4. (**A**) Absolute number of syntenic genes and of different species to which each miscanthus gene is syntenic. (**B**) Average number of syntenic relationships with species belonging to different angiosperm clades across the genes in panel A. (**C**) Absolute number of syntenic genes and of different species to which each switchgrass gene is syntenic. (**D**) Average number of syntenic relationships with species belonging to different angiosperm clades across the genes in panel C.

**Figure 9 plants-12-00779-f009:**
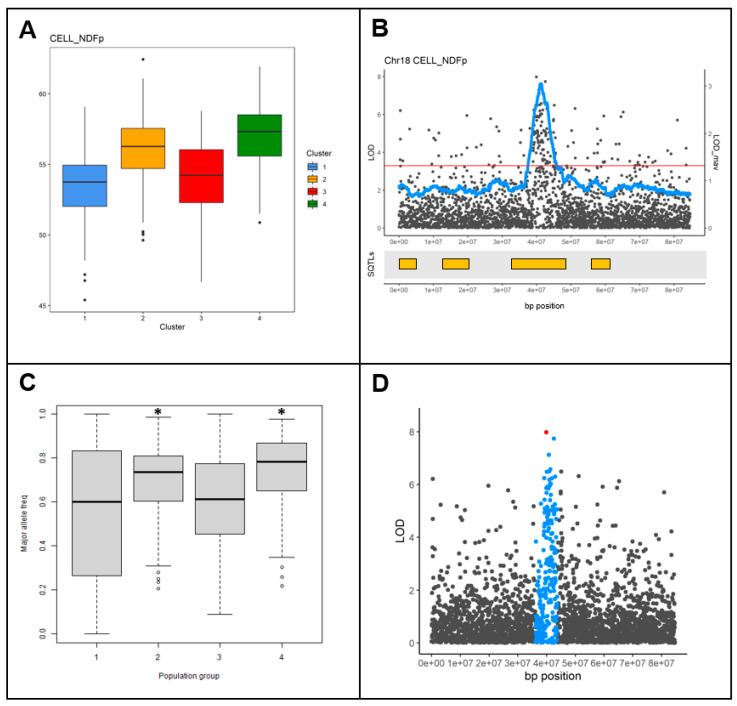
The steps followed for the detection of the “extra” GWAS QTLs. (**A**) Significant variation for one trait across the four main population structure groups of the GWAS panel. (**B**) An LOD peak from the GWAS without population structure correction for the same trait as in panel A. This peak was not included in the “standard” GWAS and colocalizes with a known SQTL region (yellow bars). (**C**) Assessment of the variation in the major allele frequency across the four population structure groups such as in panel A for all the markers included within the LOD peak of panel B. MAF differs significantly across population structure groups (asterisks denote significant differences at α = 0.05). (**D**) Identification of the “extra QTL”.

**Table 1 plants-12-00779-t001:** List of cell wall genes found within the miscanthus QTLs from the GWAS and conserved through SQTLs. The IDs of SQTL are reported as in Pancaldi, Vlegels, Rijken, van Loo, and Trindade [1].

Gene_ID	Chrom	Conserved in SQTL	Cell Wall Function	Notes	References
GQ_01G229400	Chr01	MSQTL_122	CSL	Homolog of AtCSLA9, involved in the synthesis of mannans.	[22]
GQ_01G468200	Chr01	MSQTL_122	CSL	Homolog of AtCSLA9, involved in the synthesis of mannans.	[22]
GQ_01G471400	Chr01	MSQTL_5	BLH6	Transcription factor promoting secondary cell wall synthesis in grasses	[20,23]
GQ_01G471500	Chr01	MSQTL_5	BGAL	Beta-galactosidase involved in the degradation of several polysaccharides and cell wall remodeling	[24]
GQ_01G474600	Chr01	MSQTL_122	FRA3	Homolog of AtFRA3, which coordinates actin organization during cellulose deposition	[25]
GQ_01G478800	Chr01	MSQTL_212	4CL	Important gene of the lignin synthesis pathway	[16]
GQ_03G236400	Chr03	MSQTL_216	Endoglucanase/KOR	Important genes for cellulose and plant cell wall metabolism	[26,27]
GQ_03G240500	Chr03	MSQTL_248	Endoglucanase/KOR	Important genes for cellulose and plant cell wall metabolism	[26,27]
GQ_07G169800	Chr07	MSQTL_180	CCOAOMT	Central gene for lignin synthesis, affecting both lignin amount and monolignols ratio	[16]
GQ_07G169900	Chr07	MSQTL_180	LAC	Gene involved in the in muro lignin deposition	[16]
GQ_07G174800	Chr07	MSQTL_180	MYB103	Regulates F5H expression and S-lignin deposition in arabidopsis	[28]
GQ_07G435600	Chr07	MSQTL_245	ERF39	Binds to promoters of CESA1/3/6, synthesizing primary-like cell walls	[29]
GQ_07G435700	Chr07	MSQTL_245	ERF39	Binds to promoters of CESA1/3/6, synthesizing primary-like cell walls	[29]
GQ_07G477100	Chr07	MSQTL_445	BXL	Inhibits xylan synthesis	[30]
GQ_12G086100	Chr12	MSQTL_118	XND1/WND1A	Major transcription factor in regulating secondary cell wall synthesis	[31]
GQ_12G091100	Chr12	MSQTL_118	Endoglucanase/KOR	Important genes for cellulose and plant cell wall metabolism	[26,27]
GQ_12G150500	Chr12	MSQTL_38	PAL	First step of the lignin pathway	[16]
GQ_12G153400	Chr12	MSQTL_38	UGT72E3	Gene influencing the kinetics of lignin deposition	[32]
GQ_12G158800	Chr12	MSQTL_2	VND4	Master regulator of secondary cell walls	[20,31]
GQ_12G165800	Chr12	MSQTL_91	COBL/COBRA	Homolog of brittle stalk2 locus of maize	[33]
GQ_12G168300	Chr12	MSQTL_2	WRKY12	Involved in regulating secondary cell wall and flowering in *Miscanthus lutarioriparius*	[20,34]
GQ_18G102600	Chr18	MSQTL_178	CSL	Gene involved in hemicellulose biosynthesis	[17]
GQ_18G114300	Chr18	MSQTL_185	MYB20/MYB43	MYB20, MYB43, and MYB85 regulate secondary cell wall formation	[20,31,35]

**Table 2 plants-12-00779-t002:** List of cell wall genes from the switchgrass QTLs mapped by Ali, Serba, Walker, Jenkins, Schmutz, Bhamidimarri, and Saha [21] and conserved through SQTLs. QTL traits refer to the traits associated to the QTLs from Ali, Serba, Walker, Jenkins, Schmutz, Bhamidimarri, and Saha [21] where the genes were found as retained. The SQTL IDs are reported as in Pancaldi, Vlegels, Rijken, van Loo, and Trindade [1].

Gene_ID	Chrom	QTL Trait(s) [21]	Conserved in SQTL	Cell Wall Function
IV_1KG461400	Chr_01K	Glucose; total cell wall sugar	MSQTL_195;	4CL
IV_6NG354300	Chr_06N	Total cell wall sugar	MSQTL_80;	BGAL
IV_3NG135100	Chr_03N	Glucose	MSQTL_25;	BGLU46
IV_9KG034800	Chr_09K	Xylose	MSQTL_102;	BGLU46
IV_1NG408000	Chr_01N	Total cell wall sugar	MSQTL_185;	Endoglucanase/KOR
IV_6NG323500	Chr_06N	Glucose; total cell wall sugar	MSQTL_45;	FRA1
IV_1NG306100	Chr_01N	Total cell wall sugar	MSQTL_18;	GALT/HPGT
IV_2NG448300	Chr_02N	Klason lignin	MSQTL_54;	GAUT1
IV_2KG434900	Chr_02K	Glucose	MSQTL_230;	GH17
IV_5NG130900	Chr_05N	Klason lignin	MSQTL_123;	GLCAT
IV_5NG144100	Chr_05N	Glucose; xylose	MSQTL_130;	IRX10/10L
IV_2KG435600	Chr_02K	Glucose	MSQTL_230;	LAC
IV_5KG613400	Chr_05K	Glucose	MSQTL_350;	LAC
IV_1NG407200	Chr_01N	Total cell wall sugar	MSQTL_185;	MED/REF
IV_1NG407100	Chr_01N	Total cell wall sugar	MSQTL_185;	MYB20/MYB43
IV_6NG354000	Chr_06N	Total cell wall sugar	MSQTL_80;	MYB4/MYB6/MYB7/MYB21/MYB32
IV_2NG449200	Chr_02N	Klason lignin	MSQTL_248;	MYB42/MYB85
IV_6NG352900	Chr_06N	Total cell wall sugar	MSQTL_80;	MYB42/MYB85
IV_6NG354100	Chr_06N	Total cell wall sugar	MSQTL_80;	PRX
IV_9KG292900	Chr_09N	Klason lignin	MSQTL_122;	RHM
IV_1KG460000	Chr_01K	Glucose; total cell wall sugar	MSQTL_164;	XET/XTH

## Data Availability

Not applicable.

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
