# Peer review of "Syntenic Cell Wall QTLs as Versatile Breeding Tools: Intraspecific Allelic Variability and Predictability of Biomass Quality Loci in Target Plant Species"

_plants, 2023, doi:10.3390/plants12040779_

Round 1

Reviewer 1 Report

The authors used GWAS analysis in this study. Do you think if it is better to give a brief introduction of GWAS in the paper?

In line 306, does ‘co-localized for >50% of their bp length’ mean that the sQTL locates within 0.50 times the QTL length around the QTL?

In line 635, the authors mentioned that LMM was used to conduct GWAS via statgenGWAS R package. Do you think if it is more convicing that multiple GWAS methods are used to analyze the data?

Do you think that sQTLs can be used in genomic prediction?

Author Response

Reviewer 1

The authors used GWAS analysis in this study. Do you think if it is better to give a brief introduction of GWAS in the paper?

Response: We thank the reviewer for raising this question. In our study, the GWAS is “accessory” to one the research goals, namely the testing of the co-localization of SQTLs with QTLs identified through GWAS in miscanthus, and from a previous study that performed QTL mapping in switchgrass. For this reason, we believe that the explanation given in the introduction, which mentions “association mapping” (line 92 of the revised manuscript) is sufficient to understand the research. In the case more information over the GWAS are required, paragraph 4.4 of the methods detailly describes everything about the GWAS performed.

In line 306, does ‘co-localized for >50% of their bp length’ mean that the sQTL locates within 0.50 times the QTL length around the QTL?

Response: We thank the reviewer for pointing out this request for clarification, which made us realizing that the sentence at line 306 (line 319 in the revised version) is indeed unclear. This sentence means that the 35 QTLs from the miscanthus GWAS indicated there have more than half of their region located on a genomic location where a SQTL is also present. Following the comment of the reviewer, we changed the sentence accordingly, to make it clearer. The new sentence is: “Moreover, it was observed that 35 of the 91 GWAS QTLs (38%) co-localized for >50% of their bp length with genomic regions where the miscanthus SQTLs are also present (Figure 6)” (lines 318-320 of the revised manuscript).

In line 635, the authors mentioned that LMM was used to conduct GWAS via statgenGWAS R package. Do you think if it is more convicing that multiple GWAS methods are used to analyze the data?

Response: We thank the reviewer for raising this point. StatgenGWAS is a rather novel R package which is significantly more efficient and accurate than other available packages for performing GWAS analyses, given the improved methodologies to treat the SNP and population relatedness matrices underlying GWAS models. This higher efficiency and accuracy are the reasons why this specific package was chosen for this analysis. Regarding the specific choice of a Linear mixed model (LMM), this was chosen as it allows to account for the non-independence of related individuals (random effect of population structure), in addition to modelling the fixed effects of SNP markers on phenotypic values. As shown in Supplementary Figure 10, the implementation of such LMM allowed an optimal control of population structure effect, and overall very good and convincing predictions of SNP associations with phenotypes.

Do you think that sQTLs can be used in genomic prediction?

Response: We thank the reviewer for this relevant question. We think SQTLs could be used as ancillary tools to improve genomic prediction models. However, this was not tested in this study, and as such we confined our discussion on the practical use of SQTLs to the context of association mapping via GWAS. At this point, we feel that it would be too speculative to discuss about the use of SQTLs in genomic prediction. This could be the topic for a separate research.

Reviewer 2 Report

General Observations:

The authors conducted research on Syntenic cell wall QTLs (SQTLs) to identify genetic determinants of biomass traits in 15 understudied species based on results from model crops. They also performed GWAS on cell wall quality traits on miscanthus, to verify co-localization between found loci and miscanthus SQTLs. The research is of great importance to the scientific community as this contributes to plant breeding efforts of improving quality traits. However, consider using simple sentences to avoid ambiguity. Also, avoid repeating conjunctions and try to vary them if possible e.g.  ¨to conclude¨ and ¨however¨ have been repeated many times. Overall the research is well written and I congratulate the authors for their effort.

Other concerns 

•  Consider defining ¨biomass crops¨ at the introductory part. It is also recommended to use under-domesticated crops, or understudied crops, or orphan crops but not to use them all in one draft. Give a little background on the economic importance of these biomass crops.  Aside from the addition of variability how have the results helped predict biomass quality in the studies of these plant species? 

•   Notwithstanding the future recommendations postulated, the conclusions fail to mention the current relevance of the study to breeding and how this can translate into practical use. 

Author Response

Reviewer 2

General Observations:

The authors conducted research on Syntenic cell wall QTLs (SQTLs) to identify genetic determinants of biomass traits in 15 understudied species based on results from model crops. They also performed GWAS on cell wall quality traits on miscanthus, to verify co-localization between found loci and miscanthus SQTLs. The research is of great importance to the scientific community as this contributes to plant breeding efforts of improving quality traits.

Response: We thank the reviewer for the time and effort put in reviewing our manuscript and the appreciation of our work.

However, consider using simple sentences to avoid ambiguity. Also, avoid repeating conjunctions and try to vary them if possible e.g.  ¨to conclude¨ and ¨however¨ have been repeated many times.

Response: We thank the reviewer for this comment. Following this comment, we have simplified some sentences throughout the revised manuscript, and we also replaced some of the repeated conjunctions mentioned here.

Overall the research is well written and I congratulate the authors for their effort.

Response: We thank the reviewer for the nice words.

Other concerns 

Consider defining ¨biomass crops¨ at the introductory part.

Response: We thank the reviewer for this relevant suggestion. In the revised version of the manuscript, we have added some new sentences at lines 71-78, which define biomass crops while giving an example of a type of crops which could benefit from the availability of SQTLs. This is the added piece of text: “As an example, the improvement of biomass crops – which are all the plant species that can produce biomass to sustain bio-based value-chains (Trindade et al., 2010, Mehmood et al., 2017) – could greatly benefit from this prospect. In fact, they include several under-domesticated crops (see Mehmood et al. (2017) and Pancaldi and Trindade (2020) for comprehensive lists), whose breeding cycles are highly time-consuming (Clifton‐Brown et al., 2018), while several complex plant traits should be improved in these species to allow their cultivation on marginal lands to avoid competition with food production (Pancaldi and Trindade, 2020).”

It is also recommended to use under-domesticated crops, or understudied crops, or orphan crops but not to use them all in one draft.

Response: We thank the reviewer for this comment. We decided to use only the term “under-domesticated” crop in the revised version of the manuscript. We believe this term is the most appropriate in the context of our research.

Give a little background on the economic importance of these biomass crops.

Response: We thank the reviewer for this comment. At lines 71-78 of the revised version of the manuscript we provide a definition of biomass crops, following an earlier comment provided by the reviewer. In this definition, we state that biomass crops are “all the plant species that can produce biomass to sustain bio-based value-chains (Trindade et al., 2010, Mehmood et al., 2017)”. We think this information is sufficient – in the context of our research (which has a bioinformatic-genomic technical angle) – to understand that biomass crops are economically importance without going out of scope. More detailed information on this topic can be found by the reader in some of the references reported in our article.

Aside from the addition of variability how have the results helped predict biomass quality in the studies of these plant species?

Response: We thank the reviewer for this comment. In both the Results and the Discussion session we have mentioned that the fact that SQTLs were demonstrated to co-localize with QTLs of miscanthus and switchgrass means that SQTLs can be used to map loci associated with biomass quality in novel species without performing pre-breeding steps. This can significantly save time and money in breeding programs of under-domesticated biomass crops, and constitutes a great asset to predict the location of genomic regions associated to biomass quality in novel species.

Notwithstanding the future recommendations postulated, the conclusions fail to mention the current relevance of the study to breeding and how this can translate into practical use. 

Response: We thank the reviewer for this comment. We acknowledge the importance of the point raised by the reviewer, and we think that the sentence at lines 691-694 of the revised version of the manuscript gives enough practical perspective on how SQTLs can be used in breeding activities, by stating that: “The results showed that SQTLs are valuable tools to improve breeding activities in novel crops, where they can be applied in different ways to either screen plant material for genes or alleles of interest, or to pinpoint relevant loci based on the information from model species in novel accession panels or crops.”. We think that going beyond this sentence would be too speculative, as the delineation of more specific pipelines in which SQTLs can be used would require their testing in separate experiments designed on purpose for this goal.

Reviewer 3 Report

This paper shows a very exhaustive work in which the authors did an in-depth in silico analysis of cell wall traits SQTLs for predicting genomic regions that may be potentially used in breeding programmes. So, they identified important loci potentially associated to biomass quality, the existence of intra-specific allelic variability in conserved regions and linked these findings with protein sequence modifications and the functional relevance of such modifications, based on which was already reported in the literature. Moreover, they assessed the validity of SQTLs for predicting genomic loci associated with biomass quality traits in novel plant populations and species, such as Miscanthus sinensis and Panicum virgatum.

Proposed modifications:

I Think that the sentence “This research focused on these aspects.” can be omitted.

There are a lack of references, for example in the section “2.3 Intra-specific SQTL variability leads to changes in cell wall protein sequences with a potential functional impact” as well as in table1, that should be introduced.

Line 45: “… because already proven to be causative of variability in the trait in some species” doesn’t sound good

Line 52 : delete the last –

Line 84: Miscanthus

Line 97: Replace “This figure” by “figure 2”

Figure 1 subtitle: error in line 141 “on protein sequences and strictures were also assessed”

Figure 3: lack of legend in the graphs. Poaeceae is “highlighted”

Line 454: Brachypodium distachyon in italic

Author Response

Reviewer 3

This paper shows a very exhaustive work in which the authors did an in-depth in silico analysis of cell wall traits SQTLs for predicting genomic regions that may be potentially used in breeding programmes. So, they identified important loci potentially associated to biomass quality, the existence of intra-specific allelic variability in conserved regions and linked these findings with protein sequence modifications and the functional relevance of such modifications, based on which was already reported in the literature. Moreover, they assessed the validity of SQTLs for predicting genomic loci associated with biomass quality traits in novel plant populations and species, such as Miscanthus sinensis and Panicum virgatum.

Response: We thank the reviewer for the time spent for reviewing our manuscript and the appreciation of our work.

Proposed modifications:

I Think that the sentence “This research focused on these aspects.” can be omitted.

Response: We thank the reviewer for this observation. In the revised version of the manuscript, we have removed the sentence (lines 18-19 of the revised manuscript).

There are a lack of references, for example in the section “2.3 Intra-specific SQTL variability leads to changes in cell wall protein sequences with a potential functional impact”, as well as in table1, that should be introduced.

Response: We thank the reviewer for this comment. In the revised version of the manuscript, we have included several new references that provide support for the function described for all the genes mentioned in lines 213-230 (paragraph 2.3). Moreover, we included all the references needed to support what reported in the column “Notes” of Table 1.

Line 45: “… because already proven to be causative of variability in the trait in some species” doesn’t sound good

Response: We thank the reviewer for this comment. In the revised version of the manuscript we have changed this sentence into: “On the other hand, the co-localization of syntenic regions with previously mapped QTLs ensures the relevance of the regions identified for the involvement in the trait of interest, since the occurrence of QTLs proves that genomic regions are causative of trait variability in particular species.” (lines 44-47 of the revised manuscript).

Line 52 : delete the last –

Response: We have removed the last hyphen in the revised version of the manuscript.

Line 84: Miscanthus

Response: We have implemented this change in the revised version of the manuscript (line 93).

Line 97: Replace “This figure” by “figure 2”

Response: We thank the reviewer for this keen observation. We implemented the proposed replacement in the revised version of the manuscript.

Figure 1 subtitle: error in line 141 “on protein sequences and strictures were also assessed”

Response: We thank the reviewer for this keen observation. We implemented the proposed replacement in the revised version of the manuscript.

Figure 3: lack of legend in the graphs. Poaceae is “highlighted”

Response: We thank the reviewer for this observation. We checked the correctness and corrected the spelling of “Poaceae”, while the explanation of bar colors is contained in the caption of the Figure.

Line 454: Brachypodium distachyon in italic

Response: We thank the reviewer for this detailed observation. We have implemented the proposed change in the revised version of the manuscript.